# A highly active Ni(II)-triadamantylphosphine catalyst for ultrahigh-molecular-weight polyethylene synthesis

Andrew L. Kocen[1], Maurice Brookhart[1] & Olafs Daugulis [1]

Here we report that the tri-1-adamantylphosphine-nickel complex $[Ad_3PNiBr_3]^-[Ad_3PH]^+$ upon activation with an alkylaluminoxane catalyzes the polymerization of ethylene to ultra-high-molecular-weight, nearly linear polyethylene ($M_n$ up to $1.68 \times 10^6$ g mol$^{-1}$) with initial activities reaching 3.7 million turnovers per h$^{-1}$ at 10 °C. Copolymerizations of ethylene with α-olefins such as 1-hexene and 1-octadecene, as well as tert-butyldimethyl(dec-9-en-1-yloxy) silane give the corresponding copolymers with no decrease in activity.

[1] Welch Center for Excellence in Polymer Chemistry, Department of Chemistry, University of Houston, Houston, TX 77024-5003, USA. Correspondence and requests for materials should be addressed to M.B. (email: mbrookha@central.uh.edu) or to O.D. (email: olafs@uh.edu)

Nickel complexes have been used extensively to catalyze olefin polymerizations[1–4]. Structurally, nearly all catalysts involve bidentate ligands coordinated to Ni(II) species. In 1995, late metal catalysts based on Ni(II) complexes bearing bidentate diimine ligands were shown to efficiently polymerize ethylene and α-olefins[1–3]. Subsequently, other systems capable of olefin polymerization have been reported. In addition to diimine and other [N,N]-based catalysts, nickel(II) species complexed with certain chelating [N,O], [P,O], and [N,P] ligands were shown to be effective for polymerizing and oligomerizing olefins[4–14]. Remarkably, very few [P,P] bidentate phosphine-late-transition metal complexes are capable of serving as ethylene polymerization catalysts[11–14]. Their efficiency is usually low, giving modest molecular weight polyethylene (PE) with low turnover numbers.

Predating these systems, mono-phosphine Ni(II) oligomerization and polymerization catalysts were described by Wilke and coworkers[15,16]. A diisopropyl-t-butylphosphine-nickel(II) complex was reported to dimerize propylene with an extremely high turnover rate, but a very short catalyst lifetime (Fig. 1a)[16]. Additionally, it was claimed in a review that a tri-t-butylphosphine-nickel(II) complex produces high-molecular-weight linear PE from ethylene (Fig. 1b)[15]. However, this review contains no experimental details concerning catalyst and polymer characterization. In the 1990s these types of catalysts were further described in the patent literature[17,18]. Cross-dimerization of ethylene and acrylates using a well-defined tri-cyclohexylphosphine-Ni(II) complex was demonstrated (Fig. 1c)[17]. Furthermore, a tri-t-butylphosphine/nickel(II)(acac)$_2$ system activated with modified methylaluminoxane was reported to polymerize ethylene to PE with a moderate molecular weight, $M_n$ = 6200 (Fig. 1d)[18]. Other than these isolated reports, monodentate phosphines have received no attention as ligands for late transition metal-catalyzed olefin polymerization despite their extensive use in many other catalytic applications. Many of these experiments were carried out with catalysts that were not fully characterized or were generated in situ. A complex previously claimed to be tBu$_3$PNiBr$_2$ is likely to be [tBu$_3$PNiBr$_3$]$^-$[tBu$_3$PH]$^+$ since that is the only material we obtain using the reported reaction conditions[17].

In view of the initially high turnover frequencies but low catalyst lifetimes of these monodentate phosphine complexes, we sought other bulky monodentate phosphine Ni(II) complexes that might be viable and efficient catalysts for synthesis of higher molecular weight PE. We were attracted to a recent report by Carrow and co-workers[19] of the synthesis of triadamantylphosphine, which exhibits very-high electron-donating ability for a trialkylphosphine and, at the same time, is essentially isosteric with tri-t-butylphosphine. The strong electron-donating ability of this phosphine may provide increased catalyst stability and the similarity of steric properties to tri-t-butylphosphine should potentially result in formation of high polymer from polymerization of ethylene.

We report here that the tri-1-adamantylphosphine-nickel complex, [Ad$_3$PNiBr$_3$]$^-$[Ad$_3$PH]$^+$, when exposed to alkyl aluminum activators, polymerizes ethylene to ultrahigh-molecular-weight PE ($M_n$ up to $1.68 \times 10^6$ g mol$^{-1}$) with initial activities reaching a remarkable 3.7 million turnovers per hour at 10 °C.

## Results

### Catalyst synthesis and characterization.
A reproducible synthesis of a triadamantylphosphine-nickel complex from simple and commercially available starting materials was achieved. Reaction of tri-1-adamanylphosphine (Ad$_3$P) with nickel bromide hydrate in 1,2-dimethoxyethane forms complex **6**, [Ad$_3$PNiBr$_3$]$^-$[Ad$_3$PH]$^+$, in good yield (Fig. 2). This complex exists as an air-stable green solid that decomposes over 24 h in solution. Faster decomposition is observed in halogenated alkane solvents such as dichloromethane. Single crystals suitable for X-ray analysis were grown by vapor diffusion of diethyl ether into a chlorobenzene solution of **6** at room temperature. The ORTEP diagram of **6** is depicted in Fig. 3. The structure is similar to the [tBu$_3$PNiBr$_3$]$^-$[tBu$_3$PH]$^+$ complex **7** reported by Alyea et al.[20] previously. The crystal structure shows a tetrahedral arrangement of ligands around the nickel atom. Complex **6** shows a shorter phosphorus-nickel bond than its tri-t-butylphosphine analog (2.38 vs. 2.48 Å), consistent with the stronger donor ability of Ad$_3$P. The Ni-Br bonds are slightly longer in **6** at 2.39–2.40 vs. 2.37–2.39 Å in **7**.

### Homopolymerization of ethylene by complex 6.
A short optimization study with respect to activator showed that best results for ethylene polymerization were obtained by employing 1750 equivalents (equiv.) of polymethylaluminoxane (PMAO-IP) activator (see Supplementary Tables 3-5 for details). The reactions are extremely exothermic, and the reactor has to be cooled to about

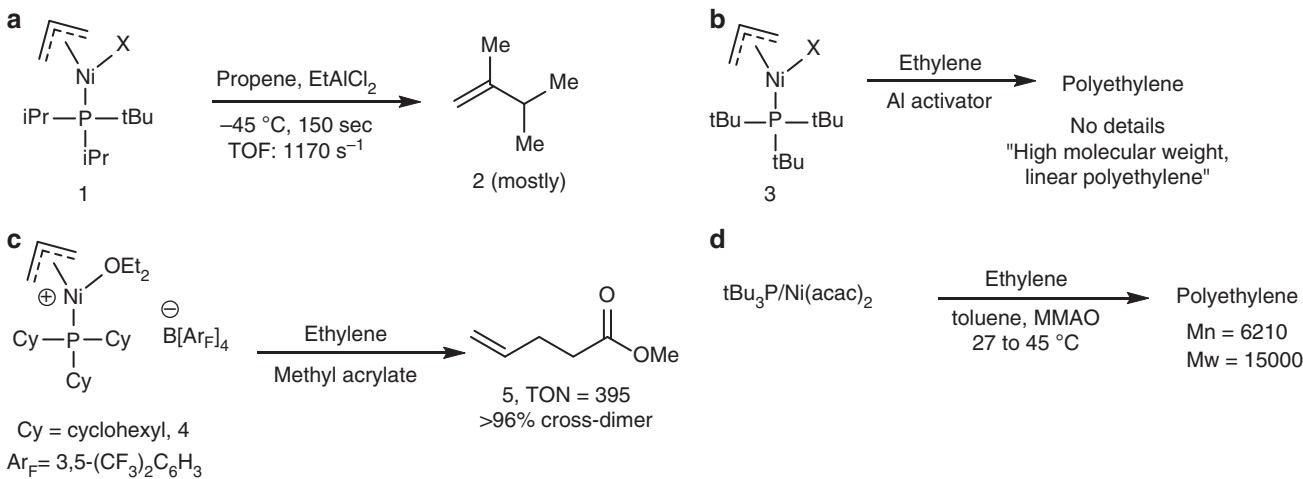

**Fig. 1** Monodentate trialkylphoshine-nickel catalysts for olefin dimerization and polymerization. **a** Propylene dimerization with exceptionally high initial turnover frequencies (Wilke). **b** Ethylene polymerization using tri-t-butylphoshine-nickel complex (Wilke). **c** Olefin cross-dimerization using tri-cyclohexylphosphine nickel catalyst. **d** Ethylene polymerization using tri-t-butylphoshine-nickel-acac catalyst generated in situ

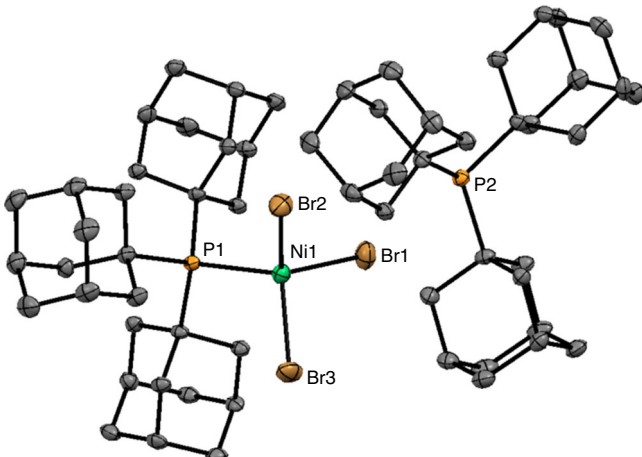

**Fig. 2** Synthesis of catalyst **6**. Reaction of tri-1-adamanylphosphine with nickel bromide hydrate in 1,2-dimethoxyethane forms complex **6**

**Fig. 3** ORTEP view of the molecular structure of **6**. Thermal ellipsoids are drawn to encompass 50% probability. Hydrogens and co-crystallized chlorobenzene are omitted for clarity. Selected bond distances (Å) and angles (deg.): P(1)-Ni(1) 2.3821(8); Ni(1)-Br(1) 2.4002(6); P(1)-Ni(1)-Br(1) 111.52 (3); Br(3)-Ni(1)-Br(2) 110.12(2)

10 °C before pressurizing with ethylene to ensure that the temperature does not rise above 25 °C. After pressurizing, the outside of the reactor must occasionally be cooled with an ice bath to maintain the desired temperature. Ethylene polymerization results are presented in Table 1. Very-high-molecular-weight, nearly linear PE is produced, with melting temperatures (second heat) in the range of 130–134 °C (see Supplementary Figures 8-11); the PE formed contains <2 branches per 1000 carbons as determined by [1]H NMR spectroscopy[10]. The formed polymer is insoluble in the reaction medium and precipitates during the reaction. Consequently, the impact of catalyst loading on turnover frequency (TOF) was analyzed to assess mass-transport issues (Table 1, entries 1–4). As the catalyst loading was decreased from 2 to 0.5 μmol, the turnover frequency increased from 120,000 to 360,000 h$^{-1}$. The molecular weight ($M_n$) of the polymer increased from 640,000 to 768,000. An increase in the solvent volume from 150 to 250 mL did not lead to an increased TOF (entry 4). Lower loading was impractical as use of catalyst quantities below 0.5μmol led to irreproducible results. These results suggest that productivity may be limited by mass-transport issues due to the insolubility of polymer.

It is difficult to carry out kinetics on such highly reactive catalysts and under conditions where the polymer precipitates; however, qualitative observations at very short reaction times at 10 °C show an approximate first-order dependence of TOF on ethylene pressure at 100 and 200 psi (Table 1, entries 6 and 10). After 3.5 min, the turnover number (TON) at 100 psi of ethylene is 79,000 (TOF = $1.4 \times 10^6$ h$^{-1}$, $M_n$ = 944,000) and doubles to 170,000 at 200 psi of ethylene (TOF = $2.8 \times 10^6$ h$^{-1}$, $M_n$ = 1,450,000). These data suggest that the catalyst resting state is a nickel alkyl complex and, given the unsaturated nature of the

complex, such a nickel alkyl species is likely to exhibit a β-agostic structure[21]. At 400 psi ethylene pressure (Table 1, entry 13), the TON increases to 220,000 with TOF = $3.7 \times 10^6$ h$^{-1}$ and $M_n$ = 1,390,000. While the TON increases relative to the 200 psi run, it does not double. This could be the result of formation at higher ethylene pressures of a second resting state, a nickel alkyl ethylene complex in equilibrium with the nickel alkyl species. Alternatively, as the molecular weight increases, mass transfer of ethylene to the active site in the precipitated polymer may be inhibited.

To further probe the catalyst behavior, we examined the polymerization at 10 °C over different times at 100 and 200 psi (Table 1, entries 6–12). At 100 psi both increase in molecular weight and TON is observed as the time increases from 3.5 to 15 min (entries 6–8). At 30 min, the TON does not increase relative to runs at 15 min (entries 8 vs. 9). It appears that the TON plateaus at about 220,000 ($M_n$ ca. 1.7 million). The same situation is observed for runs at 200 psi ethylene pressure (entries 10–12). Polymer $M_n$ reaches only ca. 1.7 million while maintaining a narrow molecular weight distribution (entry 9). The above data suggest that chain growth is greatly retarded when polymer $M_n$ values approach 1.7 million at 10 °C likely due in part to shielding of the active site in the heterogeneous polymer. However, as experiments below show, we believe catalyst decay is also an important contributing factor limiting the TON.

At 25 °C, there appears to be little impact of ethylene pressure (200 vs. 400 psi) on the TON observed in 30 min runs (Table 1, entries 3 and 5). An explanation for this observation again may be that the high $M_n$ insoluble PE produced traps and shields catalyst to prevent further chain growth. However, at 25 °C, the limiting $M_n$ values are in the range of 800,000 (vs. 1.7 million at 10 °C),

**Table 1 Homopolymerization of ethylene catalyzed by 6[a]**

| # | Ethylene pressure (psi) | Time (min) | Temperature (°C) | TON × 10$^{-4}$ | TOF × 10$^{-5}$ h$^{-1}$ | $M_w$ × 10$^{-3}$ | $M_n$ × 10$^{-3}$ | Đ |
|---|---|---|---|---|---|---|---|---|
| 1[b] | 200 | 30 | 25 | 6.2 | 1.2 | 1040 | 640 | 1.63 |
| 2[c] | 200 | 30 | 25 | 9.6 | 1.9 | 1130 | 687 | 1.64 |
| 3 | 200 | 30 | 25 | 18 | 3.6 | 1120 | 768 | 1.46 |
| 4[d] | 200 | 30 | 25 | 18 | 3.7 | 975 | 749 | 1.30 |
| 5 | 400 | 30 | 25 | 22 | 4.4 | 1360 | 793 | 1.72 |
| 6 | 100 | 3.5 | 10 | 7.9 | 14 | 1330 | 944 | 1.41 |
| 7 | 100 | 7.5 | 10 | 14 | 11 | 1820 | 1410 | 1.29 |
| 8 | 100 | 15 | 10 | 22 | 8.9 | 2340 | 1680 | 1.39 |
| 9 | 100 | 30 | 10 | 23 | 4.6 | 2270 | 1640 | 1.38 |
| 10 | 200 | 3.5 | 10 | 17 | 28 | 1980 | 1450 | 1.37 |
| 11 | 200 | 7.5 | 10 | 22 | 18 | 2030 | 1570 | 1.22 |
| 12 | 200 | 15 | 10 | 26 | 10 | 2130 | 1650 | 1.29 |
| 13 | 400 | 3.5 | 10 | 22 | 37 | 1900 | 1390 | 1.36 |
| 14 | 200 | 15 | 25 | 20 | 7.9 | 1180 | 838 | 1.41 |
| 15 | 200 | 15 | 50 | 11 | 4.4 | 600 | 411 | 1.46 |
| 16 | 200 | 15 | 75 | 2.1 | 0.85 | 239 | 171 | 1.40 |
| 17[e] | 200 | 30 | 25 | 2.8 | 0.56 | 201 | 89 | 2.26 |
| 18[f] | 200 | 7.5 | 10 | 22 | 17 | 1410 | 1100 | 1.28 |

*PMAO-IP* polymethylaluminoxane, *GPC* gel permeation chromatography, *TOF* turnover frequency, *TON* turnover number, $M_n$ number average molecular weight, $M_w$ mass average molecular weight
[a]Standard conditions: catalyst **6** 0.5 μmol, toluene 150 mL, PMAO-IP 1750 equiv, catalyst added as a CH$_2$Cl$_2$ solution, molecular weight determined by GPC in 1,2,4-trichlorobenzene
[b]Catalyst **6** 2 μmol
[c]Catalyst **6** 1 μmol
[d]Toluene 250 mL
[e][tBu$_3$PNiBr$_3$]$^-$[tBu$_3$PH]$^{+21}$ catalyst (**7**)[21]
[f][Ad$_3$PNiBr$_3$]$^-$[nBu$_4$N]$^+$ catalyst (**8**)

suggesting that catalyst decay may play a more significant role at 25 °C than at 10 °C. Broader dispersities at 25 °C support more extensive decay during polymerization. At higher temperatures, catalyst is rapidly deactivated (entries 14–16). Use of catalyst **8** possessing a tetrabutylammonium counter cation gives polymer that is similar to one obtained with **6** (entry 18).

A comparison experiment using [tBu$_3$PNiBr$_3$]$^-$[tBu$_3$PH]$^+$ **7** as catalyst (25 °C, 200 psi, entry 17) showed that both productivity (TON = 28,000) and $M_n$ (89,000) are substantially less than for **6** under the same conditions (entry 3) and a much broader dispersity, 2.26, is observed. These data clearly suggest that catalyst decay is responsible for the lower productivity of the tBu$_3$P-based catalyst. The polymer microstructure observed (ca. 2 branches per 1000 C) is similar to that obtained with **6**. The triadamantylphosphine-palladium(II) complex is inactive in ethylene polymerization (please see Supplementary Methods for details).

The reactions at 10 °C highlight two notable features of **6** as an ethylene polymerization catalyst. First, the turnover frequency of 3.7 million per hour matches the highest previously reported using any late-transition metal polymerization catalyst[3]. Second, this is a rare example of a late metal catalyst that can produce ultrahigh-molecular-weight PE (UHMWPE) with $M_n$ values reaching 1.7 million and $M_w$ values of 2.3 million[22]. Our group has reported two catalysts that produce moderately to highly branched UHMWPE[23,24]. Other groups have also reported similar materials[25–30]. By using neutral-nickel catalysts modified with pentafluorophenyl-substituted aryl groups, Kenyon and Mecking[28] have reported the synthesis of nearly linear UHMWPE by a late-transition metal complex[28].

**Copolymerization of ethylene with α-olefins**. To lower the melting temperature and decrease the crystallinity of PE branching is introduced, traditionally through copolymerization of ethylene with α-olefin comonomers[31]. We have investigated the ability of catalyst **6** to effect such copolymerizations. Results of the copolymerization of ethylene (200 psi, 25 °C) with 1-hexene are summarized in Table 2 (entries 1–4). Increasing 1-hexene concentration from 0.25 to 0.5 M results in increased incorporation of comonomer from 2.7 to 5.1 % (entries 1 and 2).

Further increase of 1-hexene concentration to 1.0 M, however, results in only a small increase in 1-hexene incorporation to 5.6 %. The molecular weight of the resulting copolymer is decreased relative to that in ethylene homopolymerization, but the TOF is somewhat increased, possibly due to a higher amorphous fraction in the copolymer and increased solvent swelling. The decrease in $M_n$ is accompanied by an increase in dispersity, suggesting that the copolymerization relative to the homopolymerization exhibits increased chain transfer relative to propagation. As in the case of the homopolymerizations, the TON plateaus around 250,000. Analysis of the copolymer by $^{13}$C NMR spectroscopy shows that only butyl branches are formed, which can result from either 1,2- or 2,1-insertions. No consecutive insertions of hexene are observed. 1-Hexene alone is not polymerized (entry 5). 1-Octadecene was also examined as a comonomer (Table 2, entries 6 and 7, 200 psi, 0.25 M comonomer). At 20 °C a remarkable TOF of 3.8 million per hour is observed for a 3.5 min run with a TON of 220,000. Increasing the polymerization time to 15 min (20 °C) results in no increase in TON. The polymerization shows compatibility with a silylated alcohol functionality. Copolymerization of ethylene (200 psi, 15 °C, 15 min) with the α-olefin *t*-butyldimethyl(dec-9-*en*-1-yloxy)silane (0.25 M) results in a TON of 230,000 and a copolymer containing 1.2 mol% comonomer. It is instructive to note that in all of the ethylene/α-olefin copolymerizations, as with the homopolymerizations, the TON is limited to ca. 220,000–240,000.

**Experiments probing catalyst stability**. As mentioned above, polymerization terminates at relatively short times. Two possible reasons include catalyst decomposition or precipitation of insoluble, high-molecular-weight polymer from solution that results in shielding of the catalyst active site. Table 3 shows catalyst stability studies at 0 °C. After combining catalyst, PMAO, and 100 equiv. of ethylene in toluene, the solution was kept for variable amounts of time (off-time). Subsequently, the flask was placed under 1 atmosphere of ethylene for 20 min. Analysis of the resulting polymer yields show that the catalyst half-life is about 25 min under these conditions. The half-life at the higher temperatures used in Table 1 will of course be shorter. Comparison of

## Table 2 Copolymerization of ethylene with α-olefins

| # | Comonomer | Concentration | Ethylene pressure (psi) | Time (min) | Temperature (°C) | TON × 10$^{-5}$ | TOF × 10$^{-5}$ h$^{-1}$ | $M_w$ × 10$^{-3}$ | $M_n$ × 10$^{-3}$ | Đ | χ (%) |
|---|---|---|---|---|---|---|---|---|---|---|---|
| 1 | 1-hexene | 0.25 M | 200 | 30 | 25 | 2.1 | 4.2 | 546 | 335 | 1.63 | 2.7 |
| 2 | 1-hexene | 0.50 M | 200 | 30 | 25 | 2.3 | 4.6 | 452 | 293 | 1.54 | 5.1 |
| 3 | 1-hexene | 1.0 M | 200 | 30 | 25 | 2.4 | 4.7 | 400 | 272 | 1.47 | 5.6 |
| 4 | 1-hexene | 0.25 M | 400 | 30 | 25 | 2.5 | 4.9 | 523 | 344 | 1.52 | 2.6 |
| 5 | 1-hexene | 0.50 M | 0 | 30 | 25 | 0 | 0 | – | – | – | – |
| 6 | 1-octadecene | 0.25 M | 200 | 3.5 | 20 | 2.2 | 38 | 535 | 374 | 1.43 | 1.9 |
| 7 | 1-octadecene | 0.25 M | 200 | 15 | 20 | 2.2 | 8.6 | 503 | 373 | 1.35 | 1.8 |
| 8 | 9D1OTBS[a] | 0.25 M | 200 | 15 | 15 | 2.3 | 9.3 | 1243 | 879 | 1.41 | 1.2[b] |

*PMAO-IP polymethylaluminoxane, GPC gel permeation chromatography, TOF turnover frequency, TON turnover number, $M_n$ number average molecular weight, $M_w$ weight average molecular weight,*
*Conditions: toluene, PMAO-IP 1750 equiv., catalyst added as a CH$_2$Cl$_2$ solution, absolute molecular weight determined by GPC in 1,2,4-trichlorobenzene, comonomer incorporation determined by*
$^{13}$C NMR
[a]*tert*-Butyldimethyl(dec-9-*en*-1-yloxy)silane
[b]Incorporation determined by $^1$H NMR

## Table 3 Decomposition study

Catalyst **6**
1) Toluene, PMAO, 100 eq C$_2$H$_4$, 0°, X min
2) C$_2$H$_4$ (1 atm), 20 min
→ Polyethylene

| # | Off-time (min) | Yield (mg) | % dead | $M_w$ × 10$^{-6}$ | $M_n$ × 10$^{-6}$ | Đ |
|---|---|---|---|---|---|---|
| 1 | 0 | 770 | 0 | 1.61 | 1.15 | 1.40 |
| 2 | 10 | 565 | 27 | 1.23 | 0.94 | 1.31 |
| 3 | 30 | 360 | 54 | 1.80 | 1.28 | 1.41 |
| 4 | 60 | 110 | 86 | 1.60 | 1.10 | 1.46 |

*PMAO-IP polymethylaluminoxane, GPC gel permeation chromatography, $M_n$ number average molecular weight, $M_w$ weight average molecular weight*
Standard conditions: Catalyst **6** 0.5 µmol, PMAO-IP 1750 equiv., toluene 25 mL, 100 equiv. ethylene added via syringe into solution; wait *X* minutes, then bubble ethylene directly into the solution. The molecular weight was determined by GPC in 1,2,4-trichlorobenzene

these data with results in Table 1 suggests that the loss of activity in polymerization is likely caused both by decomposition of catalyst and precipitation of high-molecular-weight polymer from the solution as turnover in bulk polymerization slows dramatically after 3.5 min (entry 10 vs. 11, 10 °C run). A full dataset of results can be found in Supplementary Table 1 and Supplementary Figure 3.

## Discussion

Upon activation with PMAO-IP, tri-1-adamantylphosphine-nickel complex [Ad$_3$PNiBr$_3$]$^-$[Ad$_3$PH]$^+$ polymerizes ethylene to nearly linear, ultrahigh-molecular-weight PE with $M_n$ values up to $1.7 \times 10^6$ g mol$^{-1}$. At short times, polymerization activity reaches 3.7 million turnovers per hour at 10 °C, commensurate with the most active late transition metal catalysts reported in the literature. Copolymerization of ethylene with α-olefins such as 1-hexene, 1-octadecene, and *tert*-butyldimethyl(dec-9-*en*-1-yloxy) silane gives copolymers with no decrease in activity. The combination of these features makes the catalyst unique among late metal complexes capable of olefin polymerization. The results reported here using Ad$_3$P provide impetus to explore other bulky monodentate ligands that might provide longer-lived and more efficient nickel(II)-based catalysts for ethylene polymerizations and copolymerizations.

## Methods

**Synthesis of [Ad$_3$PNiBr$_3$]$^-$[Ad$_3$PH]$^+$, 6.** In a 100 mL flame-dried Schlenk flask, nickel bromide hydrate (273 mg, 1 mmol) and tri(1-adamantyl)phosphine (436 mg, 1 mmol) were combined under nitrogen. The flask was evacuated and refilled with nitrogen three times. Dimethoxyethane (20 mL) was added and the mixture was stirred for 19 h. Diethyl ether (80 mL) was added to precipitate product as a green solid. The supernatant liquid was removed via cannula and the remaining solid was dried under vacuum. The resulting green solid was dissolved in dichloromethane and filtered through Celite® to remove excess nickel bromide. The green solution was concentrated in vacuo. The product was dissolved in chlorobenzene and then diethyl ether was added at −30 °C to recrystallize **6**. The product was filtered in air and washed with diethyl ether. The resulting green solid was dried under vacuum to give **6** (281 mg, 0.24 mmol, 48%) as a green solid. X-Ray quality crystals were grown by vapor diffusion of diethyl ether into a chlorobenzene solution of **6** at room temperature. Complex **6** decomposes slowly in solution, but is stable as a solid in air for at least 6 months.

Anal. Calcd. for C$_{60}$H$_{91}$Br$_3$NiP$_2$: C, 61.45, H, 7.82; Found: C, 61.47 H, 7.71.

**General procedure for ethylene polymerization.** A 450 mL Parr reactor with a mechanical stirrer was heated at 100 °C for at least 2 h under vacuum. After cooling to room temperature, the reactor was pressurized to the desired pressure of ethylene followed by venting. Toluene (150 mL, unless otherwise indicated) was added via cannula and the reactor was sealed. The reactor was pressurized to the desired pressure and the temperature was allowed to reach the desired value. For temperatures below 25 °C, an ice/NaCl mixture was used to cool the reactor. After the temperature reached the desired value, the reactor was vented. PMAO-IP (1750 equiv., 0.39 mL of a 7 wt% Al in toluene) was then added and solution was stirred for 5 min. A fresh solution (0.5 µmol in 0.5 mL CH$_2$Cl$_2$) of catalyst **6** in dichloromethane was added. Please note that the catalyst will decompose if kept in solution for an extended time. After that, the reactor was pressurized with ethylene and the reaction mixture stirred for the appropriate time while maintaining temperature with an external cooling bath. Subsequently, the reactor was vented and the polymer and remaining solvent were stirred with a 1:1 10% HCl/MeOH solution (100 mL) followed by filtration. The polymer was washed with methanol and acetone and dried overnight in a vacuum oven.

**Procedure for decomposition studies.** A 50 mL oven-dried Schlenk flask was cooled to 0 °C under nitrogen. Toluene (25 mL) and PMAO-IP (1750 equiv., 0.39 mL of a 7 wt% Al in toluene) were added via syringe under nitrogen. Ethylene (1.12 mL, 100 equiv.) was added via syringe directly into the solution followed by **6** (0.5 µmol, 1 equiv. in 0.5 mL CH$_2$Cl$_2$) and the mixture was stirred for the off-time. After the off-time had elapsed, ethylene was purged directly into solution with a bubbler attached to the flask to allow excess ethylene to vent. For Table 3, entry 1,

ethylene was purged directly into solution prior to catalyst addition (in place of addition of ethylene via syringe). After 20 min of stirring the ethylene purge was stopped and the flask was opened to air. The reaction mixture was quenched with a 1:1 10% HCl(aq)/MeOH solution (25 mL) and filtered. The polymer was washed with acetone and dried overnight in a vacuum oven.

## Data availability

The X-ray crystallographic data for complex **6** has been deposited at the Cambridge Crystallographic Data Center (CCDC) under the deposition number 1882446. These data can be obtained free of charge via www.ccdc.cam.ac.uk/data_request/cif. Supplementary Information contains detailed experimental procedures and characterization data for new compounds, as well as DSC (Supplementary Figures 8-11), GPC (Supplementary Figures 12 and 13), X-ray crystallography (Supplementary Table 2 and Supplementary Figure 7), and $^{13}C$ and $^{1}H$ NMR data (Supplementary Figures 1, 2, 4-6, 14-16).

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

## Acknowledgements

This work was supported by the Welch Foundation (grant E-1983 to M.B., Chair E-0044 to O.D.). We thank Dr. Megan Robertson for providing access to the DSC instrument, Tyler Cooksey for training and discussion of DSC analysis, and Dr. Xigu Wang for X-ray analysis of **6**.

## Author contributions

A.L.K., M.B. and O.D. conceived and designed the study. A.L.K. synthesized the catalyst, ran the polymerization reactions, and characterized the polymer. A.L.K., M.B. and O.D. wrote the manuscript. M.B. and O.D. directed the research.

## Additional information

**Competing interests:** The authors declare no competing interests.

