## [Peer Review File · Nature Communications]

REVIEWERS' COMMENTS:

Reviewer #1 (Remarks to the Author):

The authors have addressed my previous concerns. I think this manuscript represent an important advance in this field. I recommend the publication of this manuscript in the current form.

Reviewer #2 (Remarks to the Author):

In this work, the authors describes a bulky nickel complex ($[\text{Ad}3\text{PNiBr}3]-[\text{Ad}3\text{PH}]^+$) and its application on the production of ultrahigh-molecular-weight (M_n up to 1.68×10^6 g/mol) polyethylene with high activities (3.7 million turnovers/h). This paper is a revised version of a manuscript that I recently reviewed for Nature Catalysis. All of my concerns have been addressed, but the added results was not very satisfying. Catalytic activity of this bulky nickel catalysts and molecular weight of polyethylene both dramatically decreased (Table 1, entry 14-16). Additionally, the polymerization in presence of polar additives and the copolymerization of ethylene with acrylates was not successful. Monodentate phosphine nickel complexes have been reported for ethylene polymerization (Ref 17 and 18), and this work made a big progress by introducing a bulky phosphine group. This study is thorough and there are no technical problems but does not offer a sufficiently significant advance to warrant publication in Nature Communications.